# Comparison of Automated Machine Learning (AutoML) Tools for Epileptic Seizure Detection Using Electroencephalograms (EEG)

Swetha Lenkala [1], Revathi Marry [1], Susmitha Reddy Gopovaram [1], Tahir Cetin Akinci [2,3,*] and Oguzhan Topsakal [1]

1   Department of Computer Science, Florida Polytechnic University, Lakeland, FL 33805, USA; slenkala4630@floridapoly.edu (S.L.); rmarry4881@floridapoly.edu (R.M.); sgopavaram2200@floridapoly.edu (S.R.G.); otopsakal@floridapoly.edu (O.T.)
2   Winston Chung Global Energy Center, University of California at Riverside, Riverside, CA 92509, USA
3   Department of Electrical Engineering., Istanbul Technical University (ITU), 34469 Istanbul, Turkey
*   Correspondence: tahircetin.akinci@ucr.edu

**Abstract:** Epilepsy is a neurological disease characterized by recurrent seizures caused by abnormal electrical activity in the brain. One of the methods used to diagnose epilepsy is through electroencephalogram (EEG) analysis. EEG is a non-invasive medical test for quantifying electrical activity in the brain. Applying machine learning (ML) to EEG data for epilepsy diagnosis has the potential to be more accurate and efficient. However, expert knowledge is required to set up the ML model with correct hyperparameters. Automated machine learning (AutoML) tools aim to make ML more accessible to non-experts and automate many ML processes to create a high-performing ML model. This article explores the use of automated machine learning (AutoML) tools for diagnosing epilepsy using electroencephalogram (EEG) data. The study compares the performance of three different AutoML tools, AutoGluon, Auto-Sklearn, and Amazon Sagemaker, on three different datasets from the UC Irvine ML Repository, Bonn EEG time series dataset, and Zenodo. Performance measures used for evaluation include accuracy, F1 score, recall, and precision. The results show that all three AutoML tools were able to generate high-performing ML models for the diagnosis of epilepsy. The generated ML models perform better when the training dataset is larger in size. Amazon Sagemaker and Auto-Sklearn performed better with smaller datasets. This is the first study to compare several AutoML tools and shows that AutoML tools can be utilized to create well-performing solutions for the diagnosis of epilepsy via processing hard-to-analyze EEG timeseries data.

**Keywords:** AutoML; electroencephalogram; EEG; epilepsy; machine learning; evaluating; performance; time series

## 1. Introduction

Epilepsy is a common neurological disorder affecting up to approximately one percent of the world's population [1,2]. It is characterized by recurrent seizures, which are caused by abnormal electrical activity in the brain [3,4]. While epilepsy can occur at any age, it is more common in children and older adults [5]. The disorder is caused by abnormal electrical activity in the brain, which can be triggered by a variety of factors, including genetic predisposition, head injuries, infections, and brain tumors [6]. Epilepsy is usually diagnosed after a person has either experienced at least two unprovoked seizures (not caused by a known condition such as alcohol withdrawal or extremely low blood sugar) more than 24 h apart or when the patient experienced one unprovoked seizure with a high probability of recurrence [4]. Appropriate diagnostic tools for epilepsy include electroencephalography (EEG), computerized tomography (CT), magnetic resonance imaging (MRI), and blood tests [7].

Electroencephalography (EEG) is a widely employed, well-established, non-invasive technique for quantifying electrical activity in the brain [8,9]. It serves as a valuable

modality for acquiring informative data pertaining to brain functionality and plays a significant role in facilitating the diagnosis of epilepsy [10,11]. In comparison to alternative approaches utilized in the assessment of epilepsy, EEG devices offer portability, cost-effectiveness, and the capability to capture time-domain recordings, which can subsequently be transformed into the frequency domain [11]. Despite its inherent potential, the analysis of EEG data presents complex challenges attributable to the intricate characteristics of the signal and the substantial volume of data that necessitates thorough processing [12]. In the context of epileptic seizure detection, neurologists meticulously scrutinize and interpret information derived from EEG signals, encompassing waveform characteristics, frequency components, and amplitude attributes, given that seizures typically manifest distinctive features, notably the presence of spikes. However, achieving efficient epilepsy seizure-detection frequently entails considerable time expenditure and demands extensive efforts, both of which increase the likelihood of human error, as it relies heavily on visual inspection by a clinician. Consequently, the utilization of automated techniques emerges as an exceedingly promising prospect [7].

Time series data are a collection of sequential observations or measurements recorded over time, with the order of data points playing a crucial role in the analysis. Considering the EEG data analysis, time series data refers specifically to the recorded electrical activity originating from the brain and captured and measured at a certain time interval [13]. Time series data have distinctive features that distinguish them from other data types. In particular, temporal dependence is a key feature in which each data point is influenced by its previous observations [14]. This interdependence leads to discernible patterns and trends within the data, requiring effective capture and analysis techniques. Also, time series data often exhibit inherent features that can appear in EEG signals, such as seasonality, periodicity, or trends [15,16]. These patterns can result from natural fluctuations or systematic changes in brain activity, including circadian rhythms or neurological conditions. Understanding and extracting these natural patterns is crucial for comprehensive EEG data analysis. The use of appropriate methods for time series analysis facilitates the identification and interpretation of relevant patterns and trends in EEG data. Researchers and clinicians use these techniques to help diagnose, monitor, and understand neurological disorders [17,18].

In recent years, there has been a growing interest in using machine learning (ML) techniques to analyze EEG data [19]. ML techniques can be used to automatically extract features from the EEG signal and classify the data into different categories based on the extracted features. Applying ML to EEG data has the potential to improve the accuracy and efficiency of EEG data analysis [20]. However, applying ML to EEG data has several challenges, such as the differences in seizure morphology between patients [21], data drift (changes in the characteristics of the independent variable over time) [22,23], etc. All these challenges increase the complexity of the model and require expert knowledge to set up the ML model with correct hyperparameters. These drawbacks can be overcome by using automated machine learning (AutoML).

AutoML tools aim to make ML more accessible to non-experts and automate the ML model selection and hyperparameter tuning. AutoML tools have gained popularity in recent years and have been used to solve various ML problems. AutoML tools can be used to analyze EEG data and diagnose neurological disorders. These tools can automatically extract features from the EEG signal and classify the data into different categories, making them useful for professionals and clinicians who may not have expertise in ML or EEG data analysis [24]. Moreover, AutoML tools can reduce the time and effort required for developing an ML model and improve the accuracy of the diagnosis. Several AutoML tools are available that can be used to analyze time series EEG data. These tools use various ML algorithms, such as decision trees, random forests, and neural networks. The selection of a particular AutoML tool depends on the specific problem being addressed and the strengths and weaknesses of each tool. In this paper, we explore the use of AutoML tools for epilepsy diagnosis utilizing EEG data. We selected three leading AutoML tools and

three open-access EEG datasets. We processed the time series data in the datasets to run experiments to compare the AutoML tools' performance in diagnosing epilepsy. Overall, this paper aims to demonstrate the potential benefits of using AutoML tools for EEG data analysis and their usefulness for clinical practice.

## 2. Related Works

Ahmad et al. conducted a comprehensive review of research efforts focused on the detection of epileptic and non-epileptic signals utilizing conventional machine learning techniques [25]. Dorai and Ponnambalam proposed a hybrid model combining support vector machine (SVM) and k-nearest neighbors (KNN) for the classification of EEG epochs into categories of seizure and non-seizure [26]. Birjandtalab et al. employed a Gaussian mixture model (GMM) to detect epileptic seizures, achieving acceptable accuracy values and an F-measure of 85.1 percent [27]. In the study described by [28], artificial neural network (ANN) classifiers utilizing time-frequency domain characteristics on EEG brain activity datasets were suggested. These classifiers demonstrated a 95 percent accuracy rate in distinguishing signals as "non-seizure" or "seizure" across various EEG dataset class combinations. Satapathy et al. proposed the use of two models, SVM and neural networks, as "black box" techniques, to identify seizures in the EEG dataset. The results indicated that the SVM model outperformed other networks in terms of accuracy and time complexity [29]. Hassan and Subasi employed genetic algorithms (GA), SVM, and particle swarm optimization (PSO) to identify seizures, achieving a maximum accuracy of 92.38 percent [30]. Shoeb and Guttag employed SVM classifiers and vector characteristics on the CHB-MIT dataset, achieving a seizure prediction accuracy of 93.38 percent [31]. Amin et al. utilized discrete wavelet transform analysis along with naïve Bayes, KNN, multi-layer perceptron (MLP), and SVM classifiers, achieving an accuracy rate of 92 percent [32]. Raghu et al. implemented a hybrid KNN-SVM model on raw EEG data for precise classification of epileptic seizure detection, resulting in a testing accuracy of up to 90 percent [33]. Zabihi et al. processed their dataset using frequency-domain and time-domain features with an SVM classifier, achieving 93.78 percent sensitivity and 96.05 percent specificity [34–36]. Yuan et al. reviewed and compared traditional machine learning and recent deep learning approaches for the diagnosis of epilepsy, utilizing mostly neuroimages such as MRI, fMRI, and PET [37]. Liu et al. performed the only study that utilized one of the AutoML tools (AutoKeras) for epilepsy diagnosis using EEG data [8]. They compared the performance of AutoKeras with the manually developed model using the Bonn EEG dataset and saw that AutoKeras outperformed the manually developed ML model. To the best of our knowledge, there are no studies comparing the performance of multiple AutoML tools for epilepsy diagnosis using EEG datasets.

## 3. Methodology

In this section, we first describe how we selected the AutoML tools among the many available and the properties of these AutoML tools. Then, we explain the datasets and the performance metrics utilized in this study.

### 3.1. AutoML Tools

The AutoML tools were selected based on the following criteria (i) the AutoML framework should have capabilities specifically designed for time series analysis; (ii) the framework should have built-in pre-processing capabilities to handle EEG data, such as filtering, artifact removal, and normalization; (iii) the framework should be able to automatically select relevant features or allow for easy feature engineering; (iv) the framework should have a wide range of time series models to choose from; (v) the framework should provide robust performance evaluation metrics; (vi) the framework should be able to handle large datasets and be scalable to accommodate increasing amounts of data; (vii) the AutoML framework should be easy to use, even for non-experts, with a user-friendly interface and clear documentation. By considering these criteria, AutoML frameworks that

are best suited for time series EEG data analysis are Auto-Sklearn, Amazon Sagemaker, and AutoGluon. We next describe the details of these tools.

### 3.2. Auto-Sklearn

Auto-Sklearn is a scientifically-designed automated machine learning (AutoML) software package developed on the foundation of the scikit-learn ML library. It encompasses a diverse array of 15 classification algorithms, 14 preprocessing methods, and 4 data pre-processing techniques [38]. The ML models incorporated within this AutoML tool include gradient boosting, adaboost, random forest, K -nearest neighbors, and extra trees, which were carefully chosen for their relevance and effectiveness in various applications. Auto-Sklearn uses a meta-learning approach to assemble the best models, which involves learning the performance of different models and combining them in an optimal way. Auto-Sklearn optimizes the hyperparameters of the model using Bayesian optimization. The final stage involves assembling the best models to improve overall performance [39].

### 3.3. Amazon Sagemaker

Amazon SageMaker Autopilot was introduced by AWS [40] and is AWS's cloud-based machine learning platform. SageMaker is configured to work with popular deep-learning frameworks, including Apache MXNet, PyTorch, and TensorFlow. Amazon SageMaker Autopilot autonomously deduces the most appropriate prediction type for a given dataset, encompassing binary classification, multi-class classification, or regression. Subsequently, SageMaker Autopilot systematically explores a range of high-performing algorithms, including gradient boosting decision trees, feed forward neural networks, and logistic regression. It trains and optimizes hundreds of models based on these algorithms to find the model that offers the best fit for the data under consideration. Amazon SageMaker Autopilot effectively relieves users from the arduous task of constructing machine learning models. It automatically explores diverse solutions, aiming to identify the optimal model. Moreover, it provides an explainability report that facilitates a deeper understanding and comprehensive explanation of the models created using SageMaker Autopilot [41].

### 3.4. AutoGluon

AutoGluon is a sophisticated automated machine learning (AutoML) platform that has the capabilities of automated data pre-processing, model architecture search, and hyperparameter tuning to collectively contribute to model performance and predictive accuracy [42–44]. AutoGluon stands out due to its integration of a diverse repertoire of ML models, meticulously chosen to address a wide range of application scenarios. These models comprise weighted ensembles, lightGBM variations, extra trees variations, random forest variations, K-neighbors variations, and neural net variations. AutoGluon facilitates an efficient and systematic exploration of the most appropriate model for a given task. This adaptive selection process leads to the optimization of predictive accuracy and overall model performance, bolstering the utility and efficacy of AutoGluon as an AutoML platform [44].

The multi-layer stacking strategy used by AutoGluon involves building a stack of multiple layers of models, where the output of the first layer of models is used as input for the second layer, and so on. In each layer, a set of different types of models is used, and the outputs of all these models are concatenated to form a new feature set, which is used as input for the next layer. This approach allows higher layers to access the original data features and build more complex models. AutoGluon also innovatively applies ensemble selection in the final layers of the stack to restrain overfitting. This involves selecting the best-performing models from each layer and combining them to form an ensemble model. To further improve the performance of the stacking method, AutoGluon uses k-fold bagging, which involves splitting the training data into k subsets, training a model on each subset, and then averaging the predictions of the k models. This process is repeated multiple times to further reduce overfitting [39].

## 4. Datasets

The shortlisting criteria for datasets involved the utilization of open-access datasets comprising data from both healthy and epileptic adult patients. The selected datasets were the UC Irvine ML Repository [44], the Bonn EEG time series dataset [45], and the Zenodo dataset [46,47]. The number of features and samples in these datasets are listed in Table 1.

**Table 1.** Distribution of Samples by Class for the UC Irvine Dataset.

| Title 1 | Title 2 | Title 3 | Title 4 | Title 5 |
|---------|---------|---------|---------|---------|
| UC Irvine | 9200 | 2300 | 9200 | 11,500 |
| Bonn | 4097 | 200 | 300 | 500 |
| Zenedo | 4097 | 400 | 400 | 800 |

### 4.1. UC Irvine ML Repository

This dataset comprises five distinct classes, labeled from 1 to 5, and consists of 100 signals each, with a duration of 23.6 s. Classes 4 and 5 pertain to signals obtained from five healthy individuals, where class 4 corresponds to signals recorded with eyes closed, and class 5 corresponds to signals recorded with eyes open. The remaining three classes (1, 2, and 3) represent signals obtained from five patients diagnosed with epilepsy. Within classes 2 and 3, the recordings were captured during non-seizure periods, where class 2 was obtained from the epileptogenic region of the patient, and class 3 was recorded from the pre-seizure hippocampal contralateral hemisphere. In contrast, class 1 encompasses signals recorded during epileptic seizure episodes.

All EEG signals were captured using a 128-channel system with a sampling rate of 173.61 Hz, utilizing a 12-bit analog-to-digital converter. The dataset comprises a total of 11,500 samples, each containing 178 attributes, and exhibits a regular distribution. Notably, individuals within classes 2, 3, 4, and 5 have no history of epileptic seizures, while only individuals in class 1 have experienced such seizures. Consequently, the dataset analysis follows a binary structure, distinguishing between epileptic seizure and non-epileptic seizure instances. Classes 2, 3, 4, and 5 represent the non-epileptic cases.

Table 1 presents the distribution of samples in the dataset across different classes. Each class represents a specific condition or state of brain activity. This balanced distribution enables a fair representation of different brain states and conditions, facilitating reliable analysis and modeling for various applications in EEG data research and diagnostics [45].

To facilitate the binary classification, a new column labeled "y" has been introduced, where values of 0 denote non-epileptic cases (classes 2, 3, 4, and 5), and values of 1 denote epileptic cases. The dataset was obtained from Kaggle [45]. Individuals in classes 2, 3, 4 and 5 do not have a history of epileptic seizures, whereas discharges in individuals with a history of epileptic seizures but no seizures were analyzed in the other group. Table 2 provides an overview of the sample distribution across the five classes, with an equal number of samples allocated to each class. Specifically, there are 2300 samples representing epileptic cases, while the remaining 9200 samples correspond to non-epileptic cases.

#### 4.1.1. Bonn EEG Time Series Dataset

This database comprises 100 single channels EEG of 23.6 s with a sampling rate of 173.61 Hz. Its spectral bandwidth range is between 0.5 Hz and 85 Hz. It was taken from a 128-channel acquisition system. Five sets of EEG participants were cut out from a multi-channel EEG recording and named A, B, C, D, and E. Sets A and B are the surface EEG recorded during the eyes closed and open of healthy participants, respectively. Set C and D are the intracranial EEG recorded seizure-free from within the seizure-generating area and from outside the area of epileptic participants, respectively. Set E is the intracranial EEG of an epileptic patient during epileptic seizures. Each set contains 100 text files wherein each text file has 4097 features of 1 EEG time series in ASCII code. A bandpass filter with cut-off frequencies of 0.53 Hz and 40 Hz has been applied to the data. It is artifact-free data, and

hence no prior pre-processing is required for the classification of healthy (non-epileptic) and unhealthy (epileptic) signals. We combined all the sets into one dataset and added column y with values 1 and 0, 1 for epileptic patients and 0 for healthy patients. The total number of samples is 500 [46]. The binary digit 1 or 0 herein represents the presence or absence of epilepsy in this context.

**Table 2.** Distribution of Samples by Class for the UC Irvine Dataset.

| Class | Description | Patient State | Number of Samples | Type |
|---|---|---|---|---|
| 1 | Seizure activity is recorded from epileptic patients (EP) | General Epilepsy | 2300 | epileptic (2300 samples) |
| 2 | The tumor was observed in EP | PE (without seizures) | 2300 | |
| 3 | EEG signal was captured from a healthy brain region in EP | PE (without seizures) | 2300 | non-epileptic (9200 samples) |
| 4 | EEG signal recorded with eyes open of healthy patients | Healthy | 2300 | |
| 5 | EEG signal recorded with eyes closed of healthy patients | Healthy | 2300 | |

### 4.1.2. Zenodo EEG dataset

This dataset was generated with the motive of building a publicly available predictive epilepsy diagnosis model in 2020. All the data were taken exclusively using surface EEG electrodes for 15 healthy and epileptic patients. Training data has five healthy and five epilepsy patients. Each subject has 40 files containing single electrode EEG data. So, there are a total of 400 samples in the training data. The test data has ten healthy and ten epileptic patients. Each subject has 20 files containing single-electrode EEG data. So, there are a total of 400 samples in the test data. We combined all of the files into one file with 800 samples and added column y with values of 1 and 0, 1 for epileptic patients and 0 for healthy patients [47,48].

### 4.2. Performance Metrics

In evaluating the classification results, four performance metrics were utilized: classification accuracy, recall, precision, and F1 score. These metrics offer valuable insights into the effectiveness of the classification model.

Classification accuracy is a metric that measures the ratio of correctly classified samples to the total number of samples in a given test set. It serves as a fundamental measure to assess the overall correctness of the model's predictions. Accuracy (ACC) can be calculated by dividing the number of correctly classified samples (NCCS) by the total number of samples (TNS) (Equation (1)).

$$ACC = NCSS \ / \ TNS \tag{1}$$

In the evaluation of classification, employing both the F1 score and accuracy provides a comprehensive assessment. The F1 score incorporates precision and recall, achieving a balance between false positives and false negatives. Precision serves as a measure of accuracy in identifying positive samples, calculated by dividing the number of true positive predictions by the sum of true positive and false positive predictions. Precision assesses the ability of the model to correctly classify positive instances (Equation (2)). Similarly, recall indicates the capability of the model to accurately detect all positive samples. It is determined by the ratio of true positive predictions to the sum of true positive and false negative predictions. Recall evaluates the effectiveness of the model in capturing all relevant positive instances (Equation (3)). The F1 score (F1) combines precision (predicted positive value) and recall using the harmonic mean, providing a balanced evaluation of the model's performance (Equation (4)). It reliably quantifies the effectiveness of the classifier in

managing false positives and false negatives, thereby offering a comprehensive assessment of its classification capabilities [47–49].

$$Precission(p) = TP \ / \ (TP + FP) \tag{2}$$

$$Recall(r) = TP \ / \ (TP + FN) \tag{3}$$

$$F1 = 2Precission(p) * Recall(r) \ Precission(p) + Recall(r) \tag{4}$$

By utilizing these scientifically established performance metrics, we can rigorously evaluate the classification results in our experiment and gain a comprehensive understanding of the effectiveness and reliability of our classification model [8,47–51].

## 5. Results

The results of this study provide valuable insights into the potential of AutoML tools for the diagnosis of medical conditions. For each AutoML tool, default settings were used. We used the 80/20 train/test set split ratio. Next, we describe the results for each dataset and then show a unified comparison.

### 5.1. UC Irvine ML Repository

Table 3 presents the comprehensive performance metrics obtained from the application of various automated machine learning (AutoML) frameworks on the UC Irvine ML Repository dataset [45]. All three AutoML platforms have similar performance, with accuracy values of 0.98 and F1 scores of 0.95–0.96. However, there are slight differences in the recall and precision values. AutoGluon and Amazon Sagemaker have higher recall values (0.94–0.95) compared to Auto-Sklearn (0.93), which means they have a higher ability to identify epileptic patients. Auto-Sklearn, AutoGluon and Amazon Sagemaker have a slightly higher precision value (0.97), which means they have a higher ability to avoid incorrectly predicting epileptic patients.

**Table 3.** Performance Metrics of AutoMLs for the UC Irvine ML Repository Dataset.

| AutoML | Accuracy | F1 Score | Recall | Precision |
|---|---|---|---|---|
| AutoGluon | 0.98 | 0.96 | 0.94 | 0.97 |
| Auto-Sklearn | 0.98 | 0.95 | 0.93 | 0.97 |
| Amazon Sagemaker | 0.98 | 0.96 | 0.95 | 0.97 |

The presented performance metrics in Table 3 provide a quantitative assessment of AutoML framework capabilities and aid in comparing their relative performance on the UC Irvine ML Repository dataset. This comparison enables practitioners and researchers to make informed decisions regarding the selection and utilization of AutoML tools for analyzing similar datasets or addressing specific machine learning problems. These performance metrics offer insights into the effectiveness of each AutoML tool in analyzing the dataset to classify the patients for epilepsy diagnosis.

### 5.2. Bonn EEG Time Series Dataset

As shown in Table 4, Amazon Sagemaker has the highest accuracy value (0.93) and precision value (0.96), indicating that it has a high ability to correctly identify positive instances and avoid false positives. However, it has a lower recall value (0.90) compared to the Auto-Sklearn platform, which means it may miss some true positive instances. Auto-Sklearn has an F1 score of 0.85, which is a harmonic mean of precision and recall, indicating that it has a balanced performance between precision and recall. It has a perfect recall value of 1, which means it can correctly identify all positive instances. However, it has a lower precision value (0.74) compared to Amazon Sagemaker, which means it may have a higher number of false positive instances. AutoGluon has the lowest performance among the

three AutoML platforms, with an accuracy value of 0.76, an F1 score of 0.72, and a precision value of 0.69. It has a higher recall value (0.76) compared to Auto-Sklearn, but a lower precision value, which means it may have a higher number of false positive instances.

**Table 4.** Performance Metrics of AutoMLs for the Bonn EEG time series dataset.

| AutoML | Accuracy | F1 Score | Recall | Precision |
|---|---|---|---|---|
| AutoGluon | 0.76 | 0.72 | 0.76 | 0.69 |
| Auto-Sklearn | 0.85 | 0.95 | 1 | 0.74 |
| Amazon Sagemaker | 0.93 | 0.96 | 0.90 | 0.96 |

The performance metrics presented in Table 4 provide a quantitative assessment of the AutoML platforms' abilities to classify instances within the Bonn EEG time series dataset [46]. These metrics assist in understanding the strengths and weaknesses of each AutoML platform in analyzing and interpreting the EEG data for this particular dataset.

*5.3. Zenodo (Zen10do)*

Table 5 presents the performance metrics of the AutoML platforms for the Zenodo dataset [46,47]. Auto-Sklearn has the highest accuracy value (0.95) and F1 score (0.95), indicating that it has the highest overall performance among the three AutoML platforms. It also has a high precision value (0.96), indicating that it has a high ability to correctly identify positive instances and avoid false positives. Its recall value (0.93) is also relatively high, indicating that it can correctly identify a high proportion of true positives. Amazon Sagemaker has a lower accuracy value (0.91) and F1 score (0.88) compared to Auto-Sklearn, but still has a relatively high precision value (0.91). However, its recall value (0.85) is lower compared to Auto-Sklearn, indicating that it may miss some true positive instances. AutoGluon has the lowest accuracy value (0.81) and F1 score (0.80) among the three AutoML platforms, but it has a relatively high precision value (0.84). However, its recall value (0.76) is lower compared to the other two AutoML platforms, indicating that it may miss some true positive instances.

**Table 5.** Performance Metrics of AutoMLs for Zenodo.

| AutoML | Accuracy | F1 Score | Recall | Precision |
|---|---|---|---|---|
| AutoGluon | 0.81 | 0.80 | 0.76 | 0.84 |
| Auto-Sklearn | 0.95 | 0.95 | 0.93 | 0.96 |
| Amazon Sagemaker | 0.91 | 0.88 | 0.85 | 0.91 |

Among the three platforms, Auto-Sklearn achieves the highest performance, as evidenced by its superior accuracy, F1 score, precision, and recall values. Amazon Sagemaker demonstrates a slightly lower performance compared to Auto-Sklearn, while AutoGluon exhibits the lowest performance metrics.

*5.4. Comparison of Metrics*

The results of the AutoML comparison indicate all models generated by all AutoML tools performed well, especially with the largest dataset from UC Irvine. Overall, Amazon Sagemaker achieved the highest accuracy and precision scores across all three datasets, followed by Auto-Sklearn. However, AutoGluon produced the lowest scores among the three AutoML frameworks. In terms of F1 score and recall, all three AutoML frameworks produced competitive results, with Amazon Sagemaker performing the best for two datasets, UC Irvine and Bonn. Auto-Sklearn achieved perfect recall scores on the second dataset, indicating that it was able to correctly identify all positive cases in that dataset.

Figure 1 presents a graphical representation depicting the comparison of performance metrics for all AutoML tools across the different datasets. The metrics evaluated include accuracy, precision, recall, F1 score, and other relevant measures. The comparison aims to

assess the effectiveness and suitability of AutoML tools in the context of the specific datasets considered. The results obtained from this analysis provide insights into the strengths and weaknesses of each AutoML tool, aiding in the selection of the most appropriate tool for EEG data analysis. The graphical representation offers a concise and visual summary of the performance metrics, facilitating a comprehensive evaluation of the AutoML frameworks' capabilities and their potential impact on clinical practice.

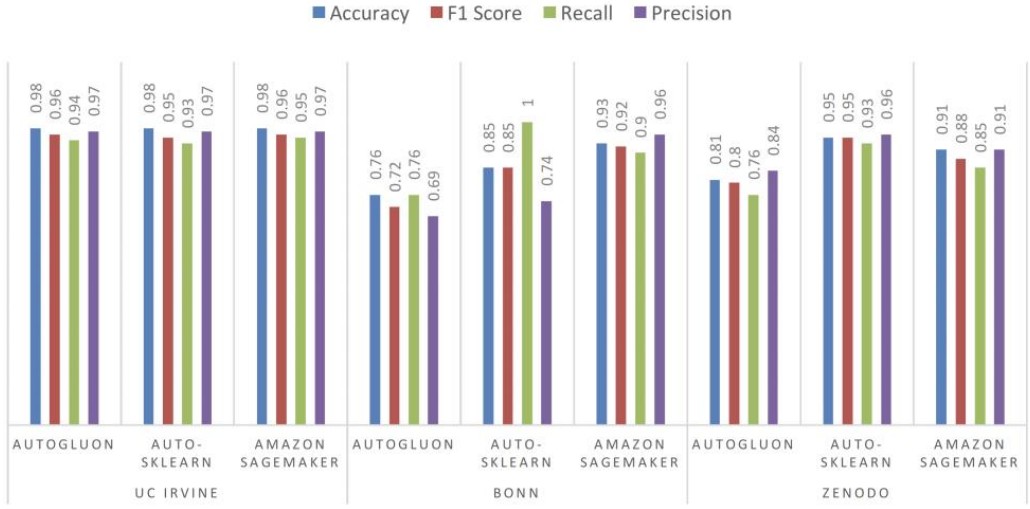

**Figure 1.** Schemes follow the same formatting.

### 5.5. Best Performing Models

To present a comprehensive evaluation of the models generated, Table 6 summarizes the best-performing models for each AutoML tool across the diverse range of datasets. An analysis of multiple datasets reveals that the WeightedEnsemble-L2 model consistently outperforms other models when utilized with AutoGluon and Amazon Sagemaker. WeightedEnsemble-L2 is a specific model type within AutoGluon and Amazon Sagemaker frameworks, and it is a type of ensemble learning model. Ensemble models are machine learning methods that combine several base models to produce one optimal predictive model. The "Weighted" part of WeightedEnsemble-L2 means that it does not just average the predictions of the base models; instead, it assigns weights to the predictions of each base model based on their performance. The idea is that models that perform better are given higher weights, and those that perform worse are given lower weights. The "L2" part refers to the method used to fit these weights. Specifically, it indicates that the weights are learned by fitting a linear regression model to the validation set predictions of the base models, with the aim of minimizing the L2 loss (also known as mean squared error). This makes the WeightedEnsemble = L2 a type of stacking ensemble, as it uses a second-level model (in this case, a linear regression) to learn how to best combine the base model predictions. The "FULL" part of the Amazon Sagemaker's models indicates that the ensemble model uses all the models generated during the process, not just a subset of them.

**Table 6.** The Best Performing Models of the AutoML Tools for the Datasets.

| Dataset | AutoGluon | Auto-Sklearn | Amazon Sagemaker |
|---|---|---|---|
| UC Irvine ML Repository | WeightedEnsemble*L2* | Gradient*Boosting* | WeightedEnsemble*L2FULL* |
| Bonn EEG time series | WeightedEnsemble*L2* | Random*Forest* | WeightedEnsemble*L2FULL* |
| Zenodo | WeightedEnsemble*L2* | K*NearestNeighbors* | WeightedEnsemble*L2FULL* |

Conversely, Auto-Sklearn demonstrates superior performance with the gradient-boosting, random-forest, and K-nearest-neighbors models when applied to the UC Irvine ML

Repository dataset, e-repository UPF, and Zenodo datasets. Gradient boosting is a powerful machine learning algorithm that creates a prediction model in the form of an ensemble of weak prediction models, typically decision trees. It builds the model in a stage-wise fashion, and it generalizes the model by allowing optimization of an arbitrary differentiable loss function. Random forest is a widely used machine learning algorithm that operates by constructing multiple decision trees during training and outputting the class by averaging the results from many individual decision trees. K-nearest neighbors (KNN) is a type of instance-based learning algorithm that works by storing all available instances and classifying new instances based on a similarity measure (such as distance functions).

## 6. Discussion

The experimental results obtained from the application of different AutoML tools on three different datasets, namely the UC Irvine ML Repository, Bonn EEG time series dataset and Zenodo are presented in Tables 3–5. These tables provide detailed performance metrics for respective AutoML tools, including accuracy, F1 score, recall, and precision. Figure 1 presents a visual comparison of the results.

Table 3 shows the performance on the UC Irvine ML Repository dataset, where all AutoML tools exhibit high accuracy and precision. For the Bonn EEG timeseries dataset, Auto-Sklearn achieved the highest accuracy and F1 score, while AutoGluon achieved the highest recall rate, as shown in Table 4. The largest dataset is UC Irvine, having 11,500 samples, and AutoML tools perform significantly better on this dataset. The performance scores are lower when used on the Bonn dataset, which has the fewest samples. Table 5 presents the results of the Zenodo dataset, where Auto-Sklearn exhibits the highest accuracy, F1 score, recall, and precision.

As shown in Table 6, the model utilizing the weighted ensemble algorithm emerged as the best-performing model for both AutoGluon and Amazon Sagemaker. Auto-Sklearn utilizes different kinds of algorithms (gradient boosting, random forest, and k-nearest neighbors) to generate its best-performing models on the datasets.

The overall findings show that AutoML tools can effectively diagnose epilepsy using EEG data. Since the models generated by AutoML tools present high accuracy and precision, they can be considered for clinical applications. However, the selection of the most appropriate AutoML tool and model depends on the particular dataset and the requirements of the application at hand. These results suggest that the choice of AutoML tool would have an impact on model performance, and researchers should carefully evaluate different AutoML tools before selecting one for their specific use case and data.

Moreover, the AutoML tools can be used for deeper analysis, such as assessing feature significance or uncovering key patterns in the data. For example, AutoGluon provides the functionality to calculate the importance of each feature used in the best model via the permutation importance technique [44]. This provides valuable insights by improving the understanding of the factors that contribute to a model's prediction.

EEG data is commonly used in rodent models to monitor the progression of epilepsy and assess experimental treatments and interventions. Several studies have investigated the use of machine learning methods to diagnose diseases in rodents using EEG data [52,53]. Yet, we have not found any research specifically employing AutoML tools for machine learning approaches with rodents. Utilization of AutoML tools can be a future task for studies that focus on rodents along with EEG data.

Data drift poses a significant challenge to core machine learning concepts, particularly maintaining consistent mean and variance. While AutoML offers a gateway for those less versed in ML to apply its principles, it does not address the data drift dilemma. AutoML users are advised to consistently monitor their models' performance and periodically employ AutoML to pinpoint the optimal model in light of new data. Established methods like stationary subspace analysis or the separation of stationary and non-stationary sources via a generalized eigenvalue problem [52,54] offer potential solutions to data drift. Future research could consider integrating these methods into AutoML platforms.

The models derived from the AutoML tools used in this study offer versatility in their use and integration with machine learning solutions or applications. After training and optimization, the best-performing models can be saved and loaded, enabling them to be deployed in different scenarios. Serialization is a viable option to preserve the structure and parameters of models, facilitating hassle-free storage and retrieval when needed. Saved models can be loaded into distribution environments where they can effectively generate predictions for previously unseen data samples. By integrating the saved models into existing systems or frameworks, real-time predictions or batch processing on new data samples can be easily performed. The specific application determines the precise use of models, including automating decision-making processes, classifying new samples, or performing regression tasks, depending on the type of model and the nature of the problem being addressed.

## 7. Conclusions

All the AutoML tools were able to generate high-performing ML models that work with three EEG datasets to diagnose epilepsy. The accuracy of the generated models ranged between 0.76 to 0.98, and the F1 score ranged between 0.72 to 0.96. All the AutoML tools performed very well, with the largest dataset achieving 98 percent accuracy and a 95–96 percent F1 score. Auto-Sklearn and Amazon Sagemaker achieved better performance scores on the smaller datasets (Bonn and Zenodo). The best-performing models for each AutoML tool for every dataset tested are listed in Table 6.

AutoML requires less human intervention for feature extraction and selecting hyperparameters and prepares the best model for the given dataset. This is the first study that evaluates AutoML tools on EEG data for epilepsy diagnosis. We show that AutoML tools can generate high-performing ML models without the need for expert ML knowledge. This study shows that AutoML tools can be utilized to create well-performing solutions for the diagnosis of epilepsy diseases via processing hard-to-analyze EEG timeseries data.

These results are important in guiding future research in selecting the appropriate AutoML tool for diagnosing epilepsy using EEG data. We evaluated three leading AutoML tools that can process EEG data. However, automated machine learning is still an emerging field, as new AutoML tools that can process EEG data become available, the potential of the AutoML tools for EEG data should be evaluated.

**Author Contributions:** Conceptualization, Methodology, and Writing—Review and Editing T.C.A. and O.T.; Investigation, Software, Writing Original Draft, S.L., R.M. and S.R.G.; validation, T.C.A., supervision, O.T. All authors have read and agreed to the published version of the manuscript.

**Funding:** This research received no external funding.

**Data Availability Statement:** Not applicable.

**Conflicts of Interest:** The authors declare no conflict of interest.

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
