# Peer review of "Comparison of Automated Machine Learning (AutoML) Tools for Epileptic Seizure Detection Using Electroencephalograms (EEG)"

_computers, doi:10.3390/computers12100197_

Round 1

Reviewer 1 Report

The paper is overall well written and clear in its exposition.

However, there are several inaccuracies.

1) Some terms are not very suitable or can be improved.

2) Table 1 is not correctly filled in.

3) Line 215: the numbering of Table 1 must be standardized and adapted to the correct editorial rules. The same problem also arises for other tables, lines: 284, 301, 318, 386, 389, 395, 397.

4 Table number 2 is not inserted correctly in the text. This also applies to table number 7.

5 Rows: 227, 242, 252 it is better to use progressive numbering i.e. 4.2, 4.3, 4.4 as done in paragraph 3: Methodology and paragraph 5: Results.

6 Line 354: same problem as point 3. Furthermore, I assume that you are referring to table number 6 and not to number 5.

The paper is overall well written and clear in its exposition.

However, there are several inaccuracies.

1) Some terms are not very suitable or can be improved.

2) Table 1 is not correctly filled in.

3) Line 215: the numbering of Table 1 must be standardized and adapted to the correct editorial rules. The same problem also arises for other tables, lines: 284, 301, 318, 386, 389, 395, 397.

4 Table number 2 is not inserted correctly in the text. This also applies to table number 7.

5 Rows: 227, 242, 252 it is better to use progressive numbering i.e. 4.2, 4.3, 4.4 as done in paragraph 3: Methodology and paragraph 5: Results.

6 Line 354: same problem as point 3. Furthermore, I assume that you are referring to table number 6 and not to number 5.

Author Response

1) Some terms are not very suitable or can be improved. [Corrected]

2) Table 1 is not correctly filled in. [Corrected]

3) Line 215: the numbering of Table 1 must be standardized and adapted to the correct editorial rules. The same problem also arises for other tables, lines: 284, 301, 318, 386, 389, 395, 397. [Corrected]

4) Table number 2 is not inserted correctly in the text. This also applies to table number 7. [Corrected]

5) Rows: 227, 242, 252 it is better to use progressive numbering i.e. 4.2, 4.3, 4.4 as done in paragraph 3: Methodology and paragraph 5: Results. [Corrected]

6) Line 354: same problem as point 3. Furthermore, I assume that you are referring to table number 6 and not to number 5. [Corrected]

Reviewer 2 Report

1. Please discuss the applicability and use of Automated Machine Learning (AutoML) Tools to Diagnose Epilepsy in rodents.  

Please improve the English in the manuscript. It requires extensive revision. 

Author Response

1) Please discuss the applicability and use of Automated Machine Learning (AutoML) Tools to Diagnose Epilepsy in rodents.  

Thank you for your kind suggestion. We have added the discussion towards the end of the ‘Discussion’ section.

2) Please improve the English in the manuscript. It requires extensive revision. 

Thank you for your kind suggestion, the paper has been checked and edited by a native English speaker.

Reviewer 3 Report

The article offers an exploration of the potential of automated machine learning (AutoML) tools in the field of epilepsy diagnosis. By leveraging AutoML tools, this research aims to enhance the accuracy and accessibility of epilepsy diagnosis through the analysis of electroencephalogram (EEG) data. The study evaluates three prominent AutoML tools, AutoGluon, Auto-Sklearn, and Amazon Sagemaker, across various datasets, ultimately shedding light on their effectiveness in this crucial medical application.

Here are my remarks:

The abstract on the portal contains line numbers. Please correct the text.

"Epilepsy is a common neurological disorder affecting approximately 1 percent of the world’s population. ""

Citation missing.

"Epilepsy is usually diagnosed after a person has either experienced at least two unprovoked seizures (not caused by a known condition such as alcohol withdrawal or extremely low blood sugar) more than 24 hours apart or when the patient experienced one unprovoked seizure with a high probability of recurrence."

Kindly provide a citation for the provided statement. It has been suggested that some forms of epilepsy may exist without the occurrence of seizures.

"data drift (changes in the characteristics of the independent variable over time) [19], etc."

The citation appears to be incorrect. Reference [19], related to the Draft Genome Sequence, does not seem to be relevant to the statement in question. I recommend reviewing and verifying the accuracy of other references for alignment with the content.

--

UC Irvine ML Repository

"Notably, individuals within classes 2, 3, 4, and 5 have no history of epileptic seizures"

What is the rationale for classifying the other classes as "normal"? Do they exhibit epileptic-like discharges?

Bonn EEG time series dataset

"We combined all the sets into one  dataset and added column y with values 1 and 0, 1 for epileptic patients and 0 for healthy  patients."

Please provide clarification regarding the combination of datasets A, B, C, D, and E. Referring to "patients" is somewhat ambiguous. Additionally, it's unclear whether some of these letters pertain to patients or exclusively represent seizure-free EEG segments (with E mainly indicating "segment free" iEEG).

Furthermore, I would suggest considering the exclusion of dataset E from the analysis, as it pertains to intracranial electroencephalography, which differs from standard EEG.

Additionally, could you provide insight into the rationale for grouping "eyes-open" and "eyes-closed" in the same category?

Zenodo EEG dataset

"We combined all files into one file with 800 samples and added column y with values of 1 and 0, 1 for epileptic patients and 0 for healthy patients.""

The duration of the segments considered is not clear (as there is no mention of the sampling rate), which leaves this information incomplete. Could you please specify whether the EEG data from epileptic patients include segments with seizures or if there are segments without seizures?

--

Valid for all datasets.

I encourage the authors to furnish additional details regarding the datasets they utilized. Specifically, it would be beneficial to explain the rationale behind the consolidation of classes and clarify whether the EEG data contains segments with seizures only (hyperexcitability, which carries an increased risk for seizures and the presence of an epileptic network within the brain. There are several types of epileptiform activity, including single discharges (sharps and spikes) and rhythmic and/or periodic activity). Additionally, providing information on the duration of the segments and resolving any ambiguity concerning whether the segments originate from age-matched individuals would be greatly appreciated.

Line 89

"We processed the time series data in the datasets to run experiments to compare the AutoML tools’ performance in diagnosing epilepsy."

I encourage the authors to furnish additional details about the processing of EEG data, including information on procedures such as filtering, resampling, and other relevant steps.

Additional Considerations:

The authors have introduced the important issue of data drift (nonstationarity), a common concern in EEG analysis. However, it remains unclear how AutoML and the tested models address this issue, which challenges fundamental machine learning principles (i.e., maintaining the mean and variance as invariant). While it is evident that this aspect can be demonstrated with the dataset used, I encourage the authors to discuss it as a potential limitation of their study and suggest avenues for future research. There are established methods, such as Stationary Subspace Analysis (as referenced in 10.1016/j.cmpb.2020.105808) or Separation of stationary and non-stationary sources using a generalized eigenvalue problem (as mentioned in 10.1016/j.neunet.2012.04.001), that could be explored in comparison to AutoML in future studies.

Furthermore, I would like to revisit the concept of epilepsy. Epilepsy is indeed a neurological disorder primarily characterized by recurrent seizures. However, it's crucial to acknowledge that not all seizures manifest with the classic convulsions that many associate with epilepsy. Some seizures can be subtle and go unnoticed by both the individual experiencing them and those in their vicinity. Additionally, not all epilepsy patients have seizures that are easily recognizable as such.

If the authors have predominantly or exclusively used EEG segments that contain seizures for the epilepsy class, I suggest considering a modification to the title. Specifically, a more appropriate title could be "Comparison of Automated Machine Learning (AutoML) Tools for Epileptic Seizure Detection Using Electroencephalograms (EEG)" or a similar variant. Diagnosing epilepsy through AutoML implies a challenging task that necessitates the incorporation of various clinical parameters alongside EEG data. I encourage the authors to reconsider and potentially adjust the title accordingly.

Author Response

  • The abstract on the portal contains line numbers. Please correct the text. The line numbers in the The abstract section is generated automatically from the Latex Template. I could not remove these line numbers.

  • "Epilepsy is a common neurological disorder affecting approximately 1 percent of the world’s population. "" Citation missing. [Corrected]

  • "Epilepsy is usually diagnosed after a person has either experienced at least two unprovoked seizures (not caused by a known condition such as alcohol withdrawal or extremely low blood sugar) more than 24 hours apart or when the patient experienced one unprovoked seizure with a high probability of recurrence."

Thank you very much, this error in the article has been corrected.

  • Kindly provide a citation for the provided statement. It has been suggested that some forms of epilepsy may exist without the occurrence of seizures.

Thank you for your suggestion, references have been added based on your kind advice.

  • "data drift (changes in the characteristics of the independent variable over time) [19], etc." The citation appears to be incorrect. Reference [19], related to the Draft Genome Sequence, does not seem to be relevant to the statement in question. I recommend reviewing and verifying the accuracy of other references for alignment with the content.

[Corrected]

  • UC Irvine ML Repository: "Notably, individuals within classes 2, 3, 4, and 5 have no history of epileptic seizures" What is the rationale for classifying the other classes as "normal"? Do they exhibit epileptic-like discharges?

Thank you very much. This uncertainty has been clarified within the paper.

  • Bonn EEG time series dataset: "We combined all the sets into one  dataset and added column y with values 1 and 0, 1 for epileptic patients and 0 for healthy patients."

The binary digit 1 or 0 herein represents the presence or absence of epilepsy in this context.

  • Please provide clarification regarding the combination of datasets A, B, C, D, and E. Referring to "patients" is somewhat ambiguous. Additionally, it's unclear whether some of these letters pertain to patients or exclusively represent seizure-free EEG segments (with E mainly indicating "segment free" iEEG).

We used the term participant to avoid confusion and corrected the text.

  • Furthermore, I would suggest considering the exclusion of dataset E from the analysis, as it pertains to intracranial electroencephalography, which differs from standard EEG.

You are correct; however, we believe that the utilization of this dataset for future analyses will add significant meaning and richness to the paper.

  • Additionally, could you provide insight into the rationale for grouping "eyes-open" and "eyes-closed" in the same category?

During the acquisition of EEG data, both open and closed-eye data are collected from individuals. This is done to allow for the investigation using both parametric and non-parametric methods. Particularly during the open and closed-eye processes, it is observable in both parametric and non-parametric methods that frequencies in the alpha band are dominant. To observe changes in the alpha band during data collection, both sets of data are collected.

  • Zenodo EEG dataset: "We combined all files into one file with 800 samples and added column y with values of 1 and 0, 1 for epileptic patients and 0 for healthy patients."" The duration of the segments considered is not clear (as there is no mention of the sampling rate), which leaves this information incomplete. Could you please specify whether the EEG data from epileptic patients include segments with seizures or if there are segments without seizures?

You are right that this information is not mentioned in the paper because there is no time-frequency analysis of the data, however, EEG data from epileptic patients includes seizure segments.

  • Valid for all datasets: I encourage the authors to furnish additional details regarding the datasets they utilized. Specifically, it would be beneficial to explain the rationale behind the consolidation of classes and clarify whether the EEG data contains segments with seizures only (hyperexcitability, which carries an increased risk for seizures and the presence of an epileptic network within the brain. There are several types of epileptiform activity, including single discharges (sharps and spikes) and rhythmic and/or periodic activity). Additionally, providing information on the duration of the segments and resolving any ambiguity concerning whether the segments originate from age-matched individuals would be greatly appreciated.

Thank you for this question. Since the data is collected according to medical ethics rules, it is clear that some of the issues you mentioned are not collected as data. Of the data collected, Epilepsy patients at the time of the visit, patients but not at the time of the visit, and non-patient participants. The data is taken from the databases and detailed information about the data is given within the references [ref 45 - 47].

  • Line 89 "We processed the time series data in the datasets to run experiments to compare the AutoML tools’ performance in diagnosing epilepsy." I encourage the authors to furnish additional details about the processing of EEG data, including information on procedures such as filtering, resampling, and other relevant steps.

[Corrected]

  • The authors have introduced the important issue of data drift (nonstationarity), a common concern in EEG analysis. However, it remains unclear how AutoML and the tested models address the data drift issue, which challenges fundamental machine learning principles (i.e., maintaining the mean and variance as invariant). While it is evident that this aspect can be demonstrated with the dataset used, I encourage the authors to discuss it as a potential limitation of their study and suggest avenues for future research. There are established methods, such as Stationary Subspace Analysis (as referenced in 10.1016/j.cmpb.2020.105808) or Separation of stationary and non-stationary sources using a generalized eigenvalue problem (as mentioned in 10.1016/j.neunet.2012.04.001), that could be explored in comparison to AutoML in future studies. Thank you very much for your suggestion to improve the manuscript. We appreciated it. We have added the following to the discussion section:

Data drift poses a significant challenge to core machine learning concepts, particularly maintaining consistent mean and variance. While AutoML offers a gateway for those less versed in ML to apply its principles, it doesn't address the data drift dilemma. AutoML users are advised to consistently monitor their models' performance and periodically employ AutoML to pinpoint the optimal model in light of new data. Established methods like Stationary Subspace Analysis [Hara] or the Separation of stationary and non-stationary sources via a generalized eigenvalue problem [Miladinović] offer potential solutions to data drift. Future research could consider integrating these methods into AutoML platforms.

  • Furthermore, I would like to revisit the concept of epilepsy. Epilepsy is indeed a neurological disorder primarily characterized by recurrent seizures. However, it's crucial to acknowledge that not all seizures manifest with the classic convulsions that many associate with epilepsy. Some seizures can be subtle and go unnoticed by both the individual experiencing them and those in their vicinity. Additionally, not all epilepsy patients have seizures that are easily recognizable as such.

[You are very right, exactly as you stated, while collecting the data, as can be seen from ref 45-47, the participants were categorized as you stated. However, this categorization is not detailed in this paper.]

  • If the authors have predominantly or exclusively used EEG segments that contain seizures for the epilepsy class, I suggest considering a modification to the title. Specifically, a more appropriate title could be "Comparison of Automated Machine Learning (AutoML) Tools for Epileptic Seizure Detection Using Electroencephalograms (EEG)" or a similar variant. Diagnosing epilepsy through AutoML implies a challenging task that necessitates the incorporation of various clinical parameters alongside EEG data. I encourage the authors to reconsider and potentially adjust the title accordingly.

[Updated]

Round 2

Reviewer 3 Report

All the issues have been effectively addressed by the authors, and the manuscript is now prepared for publication.